# Convolutional Occupancy Networks for Medical Imaging with Applications to the KiTS23 Challenge

1st Lukasz Nowakowski
Western University
London, Canada
lnowakow@uwo.ca

2nd Rajni V. Patel
Western University
London, Canada
rvpatel@uwo.ca

*Abstract*—We propose an application of occupancy networks for 3D medical image segmentation, demonstrating their effectiveness on the publicly available KiTS23 dataset. Unlike conventional CNN-based methods that operate in voxel space using encoder-decoder architectures, our approach represents anatomical structures as continuous decision boundaries within normalized coordinate space. This formulation enables fine-grained surface delineation and flexible inference resolution. Our architecture integrates a MedicalNet-pretrained ResNet encoder, a multi-scale Bi-directional Feature Pyramid Network (BiFPN) feature fusion backbone, and class-specific parallel prediction heads. To address the high anatomical variability and class imbalance in the dataset, we design a training strategy based on structured 3D patch sampling, coupled with a targeted refinement mechanism during inference that leverages coarse predictions to guide high-resolution queries for underrepresented classes. Extensive experiments show that our model achieves competitive performance on Dice and Surface Dice metrics compared to leaderboard methods. These results underscore the potential of continuous occupancy-based representations for high-fidelity medical segmentation.

*Index Terms*—Occupancy Networks, Medical Imaging, Segmentation, KiTS23 Challenge.

## I. INTRODUCTION

SEMANTIC segmentation of medical images plays a critical role in the diagnosis, treatment planning, and longitudinal monitoring of a wide range of diseases. Accurate delineation of anatomical structures and pathological regions, such as organs, tumors, and cysts, is essential for quantitative image analysis. Despite the extensive research in automated segmentation methods, manual annotation remains the clinical standard in many settings, creating a need for efficient, reliable, and generalizable segmentation models [1], [2].

In recent years, convolutional neural networks (CNNs) have emerged as the dominant paradigm in medical image segmentation [3]–[5], offering powerful nonlinear feature extractors and strong generalization across imaging modalities. Architectures based on fully convolutional networks (FCNs) and encoder-decoder designs, such as U-Net [4], have become foundational, enabling dense pixel- or voxel-level predictions while preserving contextual understanding via skip connections. These models have shown state-of-the-art performance across numerous domains, including brain [5], cardiac [6], and abdominal imaging [7].

To further enhance feature discrimination, modern architectures often incorporate attention mechanisms. Inspired by their success in computer vision and natural language processing [8], [9], attention modules have been adapted to medical segmentation tasks to emphasize salient spatial and channel-wise information [10], [11]. Models such as Attention U-Net [10], DANet [12], and PAN [13] integrate self-attention at various stages to improve long-range dependency modeling and context aggregation. Recent work has also proposed multi-scale strategies that fuse semantic information at different resolutions [14], [15], further boosting segmentation performance [16].

Despite these advancements, voxel-based models are limited by their discretized grid representations, which tie predictions to image resolution and require memory-intensive scale-fusion mechanisms to capture fine boundaries [17]–[19]. In contrast, occupancy networks learn a continuous function mapping 3D coordinates to labels [20], enabling resolution-free surface modeling and smoother, anatomically consistent segmentations.

In this paper, we explore the application of occupancy networks for 3D medical image segmentation. Our work targets the KiTS23 challenge dataset [21], focusing on the segmentation of kidneys, tumors, and cysts in CT scans (Fig. 1). The proposed model takes a $64{\times}64{\times}64$ ($[64^3]$) patch sample from the whole CT scan and uses a MedicalNet-pretrained ResNet encoder with a BiFPN head [22] to construct a canonical 3D feature grid. Query points are sampled as 3D coordinates within the normalized coordinate space $[-0.5, 0.5]^3$, representing a cube centered at the origin where each axis spans the range $[-0.5, 0.5]$. They are then processed through conditional batch normalization layers to incorporate information from the latent feature grid. These conditioned embeddings are then passed through class-specific predictor heads to estimate occupancy for each semantic class.

To improve performance across all classes and adapt to the anatomical variability present in CT imaging, we introduce structured patch sampling during training and a refinement strategy during inference. These components allow the model to handle underrepresented structures such as cysts more effectively and to iteratively refine segmentation boundaries in anatomically ambiguous regions.

Our results on the KiTS23 dataset demonstrate that this

This work was supported by a national research funding agency and institutional programs. Specific funding and affiliations are withheld for double-blind review.

Author affiliations and contact information are withheld for double-blind review.

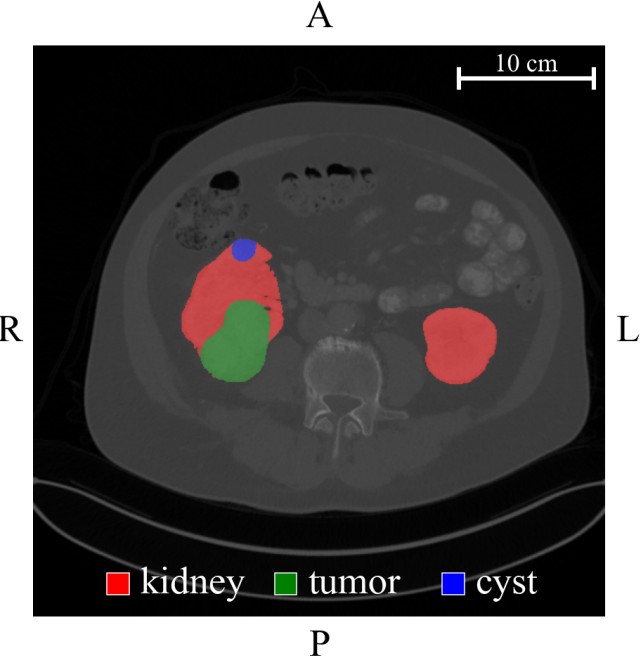

A

R                                                    L

■ kidney   ■ tumor   ■ cyst

P

Fig. 1. Example of a CT scan slice from the KiTS23 dataset, with the ground truth for the kidney, tumor, and cyst classes.

approach achieves competitive performance across Dice and Surface Dice metrics compared to voxel-based architectures, while offering the flexibility of continuous spatial reasoning and resolution-invariant boundary modeling.

## II. RELATED WORK

### A. CNN-Based Medical Image Segmentation

Convolutional Neural Networks (CNNs) have been the dominant paradigm in medical image segmentation over the past decade. U-Net [4] and its 3D extension [23] introduced a now-standard encoder-decoder structure with skip connections, enabling multi-scale feature fusion and precise boundary localization. These networks have become foundational across domains due to their simplicity, effectiveness, and extensibility. V-Net [24] introduced residual connections and a Dice-based loss function, optimizing performance in imbalanced clinical datasets. More recently, nnU-Net [25] demonstrated that careful configuration of preprocessing, architecture, and training, instead of novelty, can yield state-of-the-art performance across a wide range of biomedical tasks.

Despite their success, CNN-based models are inherently voxel-centric, predicting discrete class labels at fixed grid locations. Their reliance on pooling and upsampling operations introduces a trade-off between resolution and receptive field, and capturing fine-grained boundary information often requires architectural add-ons such as deep supervision or cascaded refinements. These models also implicitly assume that the target structures conform to the discretized spatial resolution of the input image, which may not hold for complex or small-scale anatomy.

### B. Transformer-Based Architectures in Segmentation

Transformers have recently emerged as a compelling alternative to traditional CNNs for modeling long-range dependencies. UNETR [26] replaces convolutional encoders with pure transformer blocks operating on flattened 3D patches, enabling global context aggregation from the earliest layers. Swin-Unet [27] introduces hierarchical vision transformers with window-based self-attention, achieving improved efficiency and scalability. DS-TransUNet [28] extends this idea by integrating dual Swin transformer blocks symmetrically in both encoder and decoder paths, effectively bridging local and global representations.

These transformer-based models are voxel-based at their output and typically inherit the same spatial discretization constraints as CNNs. While they excel at modeling semantic relationships across a volume, they do not fundamentally alter the representation of anatomical boundaries or the resolution of the prediction space. The segmentation outputs remain tied to a grid and suffer from the same resolution-accuracy-memory trade-offs.

### C. Occupancy Networks and Implicit Representations

Occupancy Networks [20] offer a fundamentally different representation strategy by modeling the volumetric space as a continuous function. Instead of predicting voxel labels directly, they learn a binary classifier $f : \mathbb{R}^3 \rightarrow \{0, 1\}$ that estimates whether any arbitrary point in 3D space lies inside or outside a target structure. This function is trained using point-wise supervision and conditioned on latent features extracted from the input data. Such models can represent complex shapes with sub-voxel precision and support resolution-agnostic inference, making them highly attractive for applications where boundary fidelity is critical.

Although first introduced for shape modeling and 3D reconstruction in computer graphics, occupancy networks have seen limited adoption in medical imaging. Their continuous formulation enables surface-based reasoning without committing to a discretized voxel grid, and supports mesh extraction via isosurface techniques such as marching cubes or Multi-resolution IsoSurface Extraction (MISE). Recent work such as SwIPE [29] has demonstrated the potential of implicit neural representations in segmenting whole-body CT scans, but these approaches remain rare and underexplored in the clinical domain. Our work builds upon this foundation and applies occupancy networks to a multi-class segmentation task, where each anatomical structure is predicted by an independent occupancy function conditioned on a shared latent space.

### D. Segmentation Approaches in the KiTS23 Challenge

The KiTS23 challenge presented a rigorous benchmark for kidney, tumor, and cyst segmentation in CT volumes. Among the top-performing teams, Myronenko et al. [30] used Auto3DSeg, a fully automated segmentation pipeline that integrates nnU-Net-style architectures with learned data-driven heuristics. Uhm et al. [31] proposed a multi-scale 3D U-Net combined with post-processing steps such as connected component analysis and morphological filtering to refine predictions. Other strong entries adopted ensemble strategies,

attention-based modules, or hierarchical decoders tailored for the class imbalance inherent in cyst segmentation.

All of these approaches are voxel-based and rely on dense per-voxel classification over the CT volume. By contrast, our method departs from this paradigm entirely, representing each anatomical structure as a continuous boundary function. This enables flexible querying, supports test-time resolution adaptation, and allows us to refine predictions with minimal reliance on memory-heavy volumetric inference.

## III. METHODS

In this section, we describe each component of our occupancy-based segmentation framework. We begin by outlining how query coordinates are defined and supervised, followed by an explanation of our feature encoding pipeline using a pretrained 3D ResNet and BiFPN. We then detail how query points are processed via conditional batch normalization, and conclude with descriptions of our loss function, structured patch sampling strategy, and inference-time refinement using MISE.

### A. Feature Encoding and Grid Construction

Following the preprocessing steps of [30], we rescaled the CT voxel values from $[-54, 242]$ to $[-1, 1]$, followed by a sigmoid activation function. The CT patch $x$ is processed using a 3D ResNet10 encoder pretrained via MedicalNet, which has been trained on 22 diverse clinical imaging datasets. From this backbone, we extract three spatial feature maps at resolutions $[32^3]$, $[16^3]$, and $[8^3]$. These multiscale features are passed through a Bi-directional Feature Pyramid Network (BiFPN) [22], which performs iterative top-down and bottom-up fusion, yielding a single feature grid $F_g \in \mathbb{R}^{C \times H \times W \times D}$, with channel size $C = 128$ and $H = W = D = 32$. This grid forms a dense latent representation of the input anatomy across the patch volume.

### B. Query Processing and Conditional Feature Fusion

A query point $p \in \mathbb{R}^3$ represents an arbitrary spatial location within the normalized coordinate space $[-0.5, 0.5]^3$ of a CT patch. Given a set of such points, the network evaluates their class occupancy probabilities by conditioning on local anatomical context from the CT image.

To generate a point-wise prediction, we first extract a latent feature vector $f_i \in \mathbb{R}^{128}$ by performing trilinear interpolation of the feature grid $F$ at coordinate $p_i$. This feature encodes anatomical context from the CT image in the vicinity of $p_i$. The query coordinate itself is passed through a sinusoidal positional encoding function $\gamma(p_i) \in \mathbb{R}^D$ to produce a spatially aware embedding.

These two signals, the positional embedding $\gamma(p_i)$ and the interpolated feature vector $f_i$, are fused through a sequence of five residual blocks, each containing Conditional Batch Normalization (CBN) layers. Each block applies normalization and affine transformation to the hidden features, modulated by the local context $f_i$, followed by a ReLU activation and skip connection. This produces a conditioned representation $z_i \in \mathbb{R}^{128}$ that is spatially informed and anatomically grounded. From this shared latent vector $z_i$, we apply a separate binary classifier

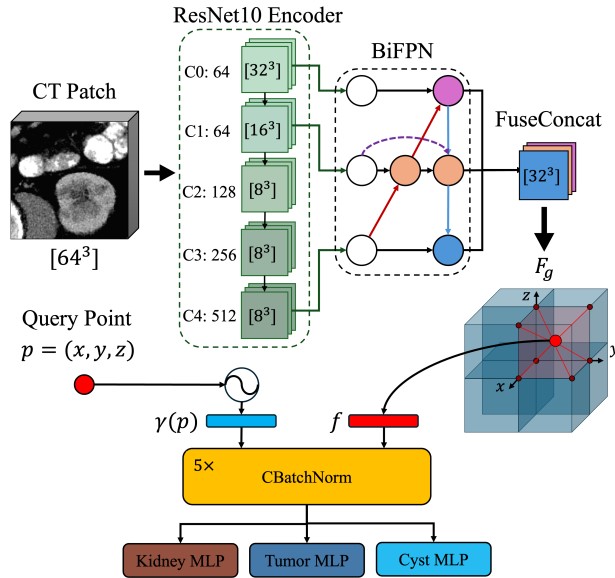

Fig. 2. Overview of the adapted ConvOccNet architecture for 3D CT scan data. The CT patch is processed through a pretrained 3D ResNet encoder producing feature maps with channel dimensions C0 → C4, followed by a BiFPN feature fusion backbone. Query points are sampled in normalized coordinate space, positionally encoded and passed through conditional batch normalization layers. The final occupancy prediction is made by a class specific MLP.

head $\mathcal{H}_c$ for each class $c \in \{1, 2, 3\}$ (kidney, tumor, cyst). Each head consists of a lightweight Multi-Layer Perceptron (MLP) (1):

$$\mathcal{H}_c(z_i) = \text{Linear}(128, 128) \rightarrow \text{ReLU} \rightarrow \text{Linear}(128, 1) \quad (1)$$

producing a final predicted occupancy value $\hat{o}_i^c = \sigma(\mathcal{H}_c(z_i))$, where $\sigma$ denotes the sigmoid activation. The complete architecture is illustrated in Fig. 2.

During inference, if all heads predict values below a predefined confidence threshold $\tau$ (i.e., $\max_c \hat{o}_i^c < \tau$), the query point $p_i$ is assigned to the background class. This allows the model to express uncertainty in ambiguous regions without forcing a foreground prediction.

### C. Training

*1) Patch Sampling:* Training is performed on 3D patches of size $[64^3]$ extracted from the original CT volumes. To ensure both anatomical diversity and balanced supervision across all classes, we employ a composite patch sampling strategy using the TorchIO library.

Each patient scan contributes ten patches per epoch:

- **Random Sampler (2 patches)**: Uniformly samples patch locations across the entire volume, without label constraints.
- **Label Sampler (4 patches)**: Prioritizes patches centered on foreground classes using per-class label probabilities, increasing the frequency of underrepresented structures.
- **Donut Sampler (4 patches)**: A custom sampler for cysts, which centers samples within a padded margin surrounding, but not inside, cyst volumes. This promotes spatial variability in cyst locations within patches and reduces positional overfitting.

TABLE I
LEADERBOARD COMPARISON ON THE KITS23 TEST SET ACROSS ALL CHALLENGE METRICS.

| Team | Avg. Dice | Avg. Surface Dice | Tumor Dice | Kidney+Masses Dice | Masses Dice | Kidney+Masses SD | Masses SD | Tumor SD |
|---|---|---|---|---|---|---|---|---|
| A. Myronenko et al. [30] | **0.835** | 0.723 | **0.758** | 0.956 | **0.792** | 0.913 | 0.641 | 0.616 |
| K. Uhm et al. [31] | 0.820 | 0.712 | 0.738 | 0.948 | 0.776 | 0.899 | 0.635 | 0.602 |
| Y. George et al. [32] | 0.819 | 0.707 | 0.713 | **0.958** | 0.785 | 0.908 | 0.640 | 0.573 |
| G. Stoica et al. [33] | 0.807 | 0.691 | 0.713 | 0.947 | 0.760 | 0.895 | 0.609 | 0.569 |
| S. Liu and B. Han [34] | 0.805 | 0.706 | 0.697 | 0.952 | 0.767 | 0.919 | 0.631 | 0.568 |
| L. Qian et al. [35] | 0.801 | 0.680 | 0.687 | 0.948 | 0.767 | 0.891 | 0.612 | 0.538 |
| Z. Salahuddin et al. [36] | 0.795 | 0.681 | 0.690 | 0.940 | 0.754 | 0.887 | 0.603 | 0.552 |
| Z. Huang et al. [37] | 0.794 | 0.692 | 0.686 | 0.951 | 0.746 | 0.909 | 0.612 | 0.556 |
| C. Chen and R. Zhang [38] | 0.799 | 0.676 | 0.691 | 0.954 | 0.752 | 0.897 | 0.591 | 0.541 |
| J. Michaud et al. [39] | 0.790 | 0.678 | 0.670 | 0.949 | 0.750 | 0.899 | 0.603 | 0.531 |
| ConvOccNet (Ours) | 0.791 | **0.744** | 0.693 | 0.947 | 0.725 | **0.930** | **0.671** | **0.652** |

This combined sampling policy significantly improves generalization and boosts minority-class performance, particularly for cyst segmentation.

*2) Point Sampling:* Within each patch, we randomly sample $N = 2048$ query points in normalized coordinate space $[-0.5, 0.5]^3$. Each point is mapped to the corresponding index in the full-resolution label mask to determine its binary label for each of the three foreground classes.

During training, points are sampled independently of their proximity to anatomical boundaries or foreground structures. This provides a broad sampling distribution over space and allows the network to implicitly learn to separate semantic boundaries through continuous occupancy classification.

*3) Loss Function:* We supervise each class prediction head using a Binary Cross-Entropy (BCE) loss (2). Let $\hat{o}_i^c$ denote the predicted occupancy value for point $p_i$ and class $c$, and $y_i^c \in \{0, 1\}$ its ground-truth label. Let $N$ be the number of query points in a patch sample. The total training loss is computed as:

$$\mathcal{L}_{\text{BCE}} = \sum_{c=1}^{3} \sum_{i=1}^{N} \text{BCE}(\hat{o}_i^c, y_i^c) \tag{2}$$

Each class head is trained independently without inter-class exclusivity. As such, query points may be predicted as foreground by multiple heads (though this is rare in practice), or assigned to background when all outputs fall below a fixed confidence threshold (as described during inference).

*4) Training Hyperparameters:* We train the model using the Adam optimizer with an initial learning rate of $1 \times 10^{-4}$, decayed by a factor of 0.75 every 20 epochs. Training is run for a maximum of 500 epochs with a batch size of 16, selected based on GPU memory constraints.

### D. Inference and Refinement

To generate full-volume predictions, we tile the CT scan using a minimal-overlap grid sampler, producing a set of $64^3$ patches that cover the entire image space. For each patch, a set of spatial query points is evaluated using our trained occupancy network, and the results are accumulated to form a unified segmentation volume.

We employ Multi-resolution IsoSurface Extraction (MISE) [20] as the backbone of our inference process. In the first pass, a coarse 3D grid of query points is evaluated across

each patch, producing class-wise occupancy scores at low resolution. MISE identifies voxels near decision boundaries by comparing occupancy values of adjacent vertices, and recursively subdivides these boundary voxels to query finer points until a target resolution is reached. Once complete, the final occupancy field is binarized and passed to the Marching Cubes algorithm to extract a continuous isosurface for each class.

To improve cyst prediction quality, we perform a second refinement pass based on the results of the first. Connected components in the initial cyst mask are identified, and new $64^3$ patches are extracted from the CT volume, centered on each component. MISE is rerun from these new locations, starting again from the coarse resolution and proceeding with its standard subdivision strategy.

This refinement aligns inference with training dynamics, as the donut sampler avoids strictly centered cysts but instead places them within a narrow padded margin, keeping them roughly central within patches without introducing positional bias.

For each query point visited in both passes, we retain the prediction with the higher class-specific sigmoid confidence score. This selective fusion ensures that refined predictions are only adopted when beneficial, while preserving coarse predictions elsewhere.

## IV. RESULTS

We evaluate our model on the publicly available KiTS23 challenge dataset using the official evaluation metrics: Dice and Surface Dice scores. Table I summarizes the average and per-class performance of our model compared to the top 10 submissions on the leaderboard.

Our model achieves a competitive average Dice of 0.791 and the highest reported average Surface Dice of 0.744 among the top leaderboard entries. These results reflect the effectiveness of our continuous occupancy formulation across multiple anatomical structures.

### A. Per-Class and Per-Region Performance

Table II presents a detailed breakdown of per-class performance for our method. The kidney class achieved the highest Dice and Surface Dice scores at 0.927 and 0.905, respectively, owing to its relatively large volume and consistent appearance across patients. Tumor segmentation yielded a Dice of 0.693

TABLE II
PER-CLASS DICE AND SURFACE DICE (SD) SCORES.

| Class | Dice | Surface Dice (SD) |
|---|---|---|
| Kidney | 0.927 | 0.905 |
| Tumor | 0.693 | 0.652 |
| Cyst | 0.447 | 0.781 |
| Kidney + Masses | 0.947 | 0.930 |
| Masses | 0.725 | 0.671 |

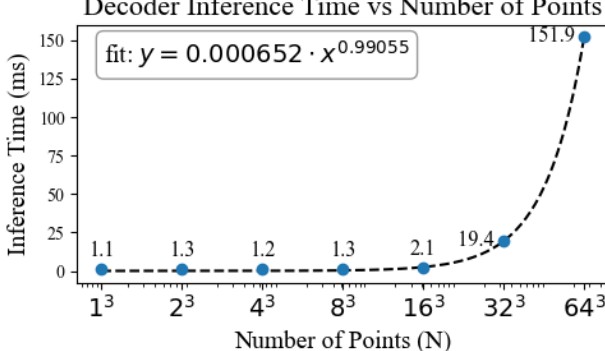

Decoder Inference Time vs Number of Points

fit: $y = 0.000652 \cdot x^{0.99055}$

Fig. 3. Decoding time (ms) as a function of the number of query points.

failure mode in which a cyst was misclassified as a tumor.

and Surface Dice of 0.652, while cysts yielded a lower Dice score of 0.447 but 0.781 for Surface Dice. These patterns reflect the inherent difficulties occupancy networks face when segmenting smaller, irregularly shaped structures.

### B. Model Efficiency and Runtime Analysis

Our model has 36.98M parameters and uses 141.07MB of memory. The encoder processes each patch in 9.40 ms on average. Decoder speed as a function of query points is shown in Fig. 3. On an NVIDIA RTX 4070 GPU, total inference takes about 6.2 seconds per CT volume without cysts, and up to 7.9 seconds with cysts. The MISE-based refinement adaptively increases resolution only near boundaries, avoiding unnecessary high-resolution queries across the whole volume.

## V. DISCUSSION

Our findings demonstrate the viability and benefits of applying occupancy networks to 3D medical image segmentation. In this section, we discuss the results, design choices, and broader implications of our work.

### A. Analysis of Results

The results in Table I confirm that our approach achieves strong performance across all evaluation metrics, with the highest reported average Surface Dice of 0.744. In particular, the kidney class attained a Dice score of 0.927 and Surface Dice of 0.905, as shown in Table II. These high scores reflect the relatively large volume and consistent appearance of the kidney across patients. Tumors, which are typically smaller and more structurally irregular, yielded lower scores (Dice 0.693, SD 0.652), while cysts, being even smaller and often indistinct, achieved a Dice of 0.447 and Surface Dice of 0.781. Figure 4 illustrates qualitative results across six cases, showing consistently high-fidelity segmentations and highlighting a

### B. Class-Specific Patterns and Sampling Implications

A closer look at the per-class results reveals that anatomical scale plays a central role in segmentation quality within occupancy-based models. The kidney class achieved the highest Dice and Surface Dice scores, reflecting its large physical size and consistent appearance across patients. With high spatial occupancy and frequent intersection by randomly sampled points, kidneys naturally provide dense supervision signals during training.

In contrast, the tumor and cyst classes occupy substantially less spatial volume and are thus more easily overlooked during patch and point sampling. Although their overall shapes are not particularly irregular, their compact size means they are statistically underrepresented within the continuous coordinate space. In the absence of targeted sampling strategies, a randomly drawn batch of 2048 points might include only a handful from a cyst, insufficient to meaningfully contribute to learning. This sparsity is exacerbated by class imbalance and further complicated by the fact that the smaller a structure is, the more its representation becomes sensitive to how and where sampling occurs.

To mitigate this, the label-aware sampler increased the occurrence of patches centered on less frequent classes, while the custom donut sampler proved particularly effective in improving cyst segmentation. By sampling the region surrounding the cyst rather than directly on it, we achieved two goals simultaneously: increased exposure of cyst voxels across training and greater variation in their spatial presentation within patches.

These adaptations underscore an important nuance in using occupancy networks for segmentation: the visibility of a structure during training is a function not only of its class frequency but also of its geometric footprint in coordinate space. Our results suggest that tailoring sampling schemes to anatomical scale can be more impactful than modifying loss functions or network capacity alone.

### C. Architectural Insights

Several architectural modifications were instrumental in achieving the presented results beyond the classical convolutional occupancy network. The use of multi-head binary predictors proved essential in alleviating class imbalance, as each class received dedicated supervision and gradient flow. Without this design, smaller classes like cysts were consistently overwhelmed by dominant structures. Although we experimented with a weighted loss to further boost sensitivity to cysts, this did not yield improvements. The multi-head design, however, proved robust and clean, especially in the context of implicit representations.

A core strength of the occupancy framework is its ability to decouple resolution from output representation. This enables dense, high-fidelity boundaries without incurring the memory costs of dense volumetric decoding. Additionally, inference-time querying allows for targeted resolution increases without

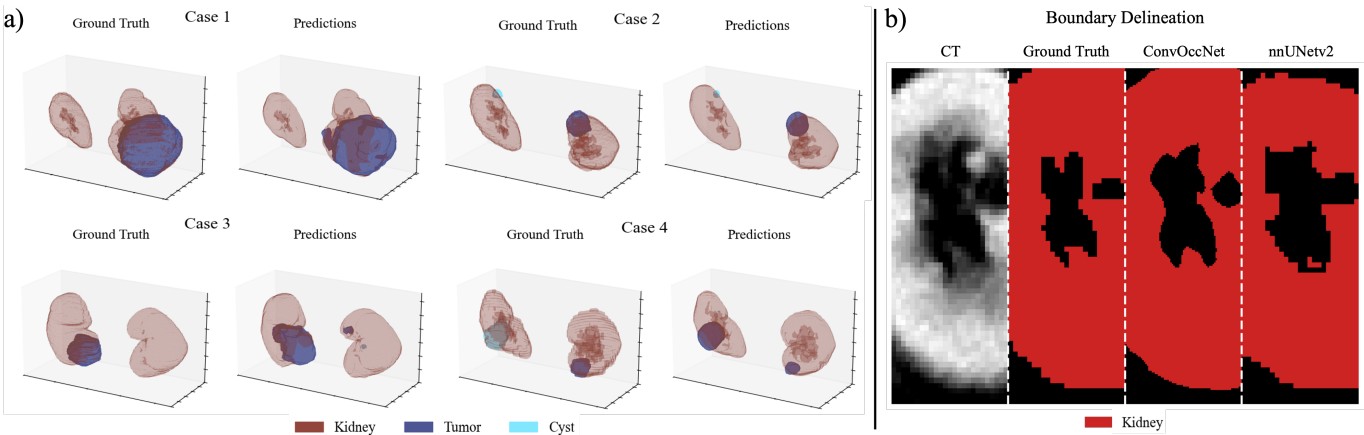

Fig. 4. a) The ground truth and ConvOccNet predicted segmentation for the kidney, tumor, and cyst classes are shown for six test cases using our two-step refinement process. b) A comparison of boundary delineation between the ground truth, ConvOccNet, and nnUnetv2 [31]. The ConvOccNet model was queried to have 0.33mm isotropic resolution, while nnUNetv2 was queried at 1mm, matching the initial scan's resolution.

architectural modification, a unique benefit not easily available to voxel-label methods.

### D. Training Lessons and Spatial Dynamics

Beyond anatomical class imbalance, our experiments revealed that the spatial distribution of occupancy points has a significant effect on training dynamics. Uniform random sampling, though simple, offered the most stable convergence due to its even and unbiased coverage of the full coordinate space. In contrast, alternative sampling methods such as near-surface sampling or cluster-weighted sampling for localized structures introduced strong spatial biases. Figure 5 illustrates the effect of querying dense regions around some classes while under-sampling others. As a result, certain regions of the coordinate space remained underrepresented throughout training, making the model sensitive to new test-time structures.

We also observed that patch size plays a subtle but important role in spatial reasoning. While the $64^3$ patch size enabled efficient training, its limited field of view sometimes excluded relevant anatomical context, particularly for tumor and cyst prediction. For instance, we observed cases where a tumor-like texture appeared ambiguous when cropped in isolation but was disambiguated in larger patches due to adjacent structures. In early overfitting experiments on single CT volumes, we found that larger patches (e.g., $128^3$) consistently produced cleaner and more stable predictions, especially in resolving tumor/cyst boundaries.

Finally, we note that training with CBN layers introduces additional sensitivity to batch size. These layers rely on accurate batch-level statistics, and small batch sizes make it difficult to compute reliable estimates of mean and variance. This amplifies early training noise and slows convergence. Larger batch sizes smooth this effect and lead to more stable function learning, though they require significantly more memory during training.

### E. Future Directions

While this work presents a strong case for implicit segmentation in volumetric data, several avenues remain for exploration. One promising direction is replacing Conditional Batch Normalization with attention-based modules. Additionally, we are actively investigating the flexibility and scalability of the architecture for use in real-time applications.

## VI. CONCLUSION

This work presented a novel application of occupancy networks to the task of 3D medical image segmentation. By representing anatomical boundaries as continuous implicit functions, our method departs from conventional voxel-based approaches and achieves highly accurate, surface-aware predictions. We introduced a hybrid architecture with multi-head binary predictors and a targeted sampling strategy that specifically addresses the challenges posed by small, under-represented structures such as cysts.

Our results on the KiTS23 dataset demonstrated that implicit representations can not only match, but in some cases exceed, the performance of state-of-the-art voxel-based models, particularly in surface accuracy and adaptability to sparse regions. More broadly, our findings highlight the importance of anatomical sampling design in the context of occupancy-based learning. We believe this work lays the foundation for further exploration of continuous segmentation models in medical imaging and opens new directions for adaptive and high-fidelity anatomical reconstruction.

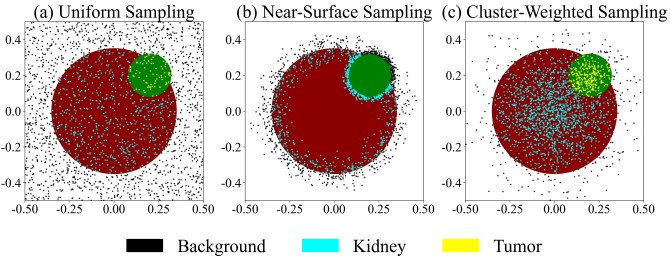

Fig. 5. Uniform, Near-Surface, and Cluster-Weighted sampling are shown in an example 2D coordinate space.

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
