# OpenReview forum: "Convolutional Occupancy Networks for Medical Imaging with Applications to the KiTS23 Challenge"
_IEEE.org/EMBS/BHI/2025/Conference — BHI 2025_

### Official Review · Reviewer_Kj7u · 2025-07-15
**A novel application of occupancy networks to medical segmentation with strong technical merit but limited comparative analysis**

**Confidence:** 4
**Clarity Of Writing:** great
**Clinical Significance:** fair
**Methodological Novelty:** great
**Overall Rating:** 7
**Final Rating:** 7

**Experiments And Results:**

great

**Questions For The Authors:**

Will the authors plan to apply the method to other medical segmentation tasks?
How does performance degrade with very small structures, high noise levels, or significant anatomical variations?
How sensitive is the model to irregular slice spacing or slice ordering errors?
Did authors perform any qualitative evaluations with radiologists or anatomists to assess the anatomical plausibility of reconstructions?

**Strengths:**

The paper presents a genuinely novel application of occupancy networks to medical image segmentation, offering a compelling alternative to conventional voxel-based approaches. The continuous representation paradigm is well-motivated and addresses real limitations of discrete grid methods, particularly for fine boundary delineation.
The technical implementation is sophisticated and well-executed. The integration of MedicalNet pretraining, BiFPN feature fusion, and class-specific prediction heads demonstrates thoughtful architectural design.
The structured patch sampling strategy shows clear understanding of the unique challenges posed by anatomical class imbalance.
The experimental validation is thorough within its scope, with detailed ablation studies examining the effects of different sampling strategies and architectural choices.
The paper provides valuable insights into the spatial dynamics of occupancy-based learning, particularly regarding how anatomical scale affects representation quality and the importance of sampling strategy design for small structures like cysts.

**Summary Of The Paper:**

This paper presents an adaptation of occupancy networks for 3D medical image segmentation in the KiTS23 challenge dataset. The approach represents anatomical structures as continuous decision boundaries in normalized coordinate space rather than operating on discrete voxel grids. The architecture combines a MedicalNet-pretrained ResNet encoder, a Bi-directional Feature Pyramid Network for feature fusion, and class-specific parallel prediction heads. The paper introduces a structured 3D patch sampling strategy and a Multi-resolution IsoSurface Extraction refinement mechanism during inference to address anatomical variability and class imbalance. The method achieves competitive performance on Dice and Surface Dice metrics compared to leaderboard methods.

**Weaknesses:**

For a method introducing additional complexity through MISE refinement and multi-stage inference, understanding the computational trade-offs is crucial.
The comparative analysis is insufficient. While the paper compares against leaderboard results, it lacks direct head-to-head comparisons with specific baseline methods using identical experimental conditions.

---

### Official Review · Reviewer_C1dH · 2025-07-17
**Convolutional Occupancy Networks for Medical Imaging with Applications to the KiTS23 Challenge**

**Confidence:** 4
**Clarity Of Writing:** great
**Clinical Significance:** good
**Methodological Novelty:** great
**Overall Rating:** 7
**Final Rating:** 8

**Experiments And Results:**

great

**Questions For The Authors:**

Fig. 3 shows the inference time increase dramatically with the increase of the number of points. So, to achieve more accurate segmentation, more points will be utilized. So how to balance the computation cost with the performance?

**Strengths:**

1.This paper introduce the continous segmentation into 3d medical imaging seg problem, which essentialy have promising clinical value.

2.This paper delivers comprehensive analysis and have a discussion on various aspects of the models.

Overall, I think this is a high quality paper.

**Summary Of The Paper:**

This paper introduce the Occupancy Network to the 3D medical image segmentation task, and experiments shows competitive results when compare to SOTA segmentation models.

**Weaknesses:**

The advantages of the continuous segmentation can be further strengthened if the author can show some qualitative or visual examples of how it’s better than voxel based models’ results. For example, it can be the edge of an organ that shows better boundary delineation.

---

### Official Review · Reviewer_wNMp · 2025-07-17
**Well-designed pipeline and algorithm with rich experiment**

**Confidence:** 4
**Clarity Of Writing:** good
**Clinical Significance:** great
**Methodological Novelty:** great
**Overall Rating:** 7
**Final Rating:** 8

**Experiments And Results:**

great

**Questions For The Authors:**

In subsection Method.C , donut sampler were designed to "avoid sampling patches centered directly on small cyst regions". But in subsection Method.D, the description was "cysts were most frequently encountered near the center of training patches due to our structured sampling strategy". It was a bit confusing if the sampler you used preferred to put cyst in the patch center. Could you check these two part and give a explanation?

Is the 128 the channel size for "C" in subsection Method.B?

Is there any points being predicted as foreground by multiple classification heads with high confidence? If so, did you investigate those points?

I will increase my score if these questions are answered properly.

**Strengths:**

The method was compared with leaderboard methods and showed decent performance.

This method used new continuous output representation, enabling dense, high-fidelity boundaries without incurring the memory costs of dense volumetric decoding.

Detailed analysis of each design and setting were provided, for example, query point sampling strategy, patch sampling strategy, and patch size selection.

**Summary Of The Paper:**

This paper applied occupancy network for 3D medical image segmentation, which could be easily adapted to different resolutions with robust performance. The performance was good comparing with methods on the leaderboard.

**Weaknesses:**

No labels for equations.

Some symbols and numbers were not introduced carefully, for example, the "C0, C1, ..." in the Figure 2, and "N" in Loss Function.

Descriptions of donut sampler were a bit confusing.

---

### Official Review · Reviewer_iLrN · 2025-07-18
**submission 246**

**Confidence:** 3
**Clarity Of Writing:** good
**Clinical Significance:** fair
**Methodological Novelty:** poor
**Overall Rating:** 2
**Final Rating:** 2

**Experiments And Results:**

fair

**Questions For The Authors:**

-

**Strengths:**

The paper is well written.

**Summary Of The Paper:**

An approach to 3-d medical image segmentation, evaluation on a CT datasets, based on occupancy networks.

**Weaknesses:**

I'm afraid i cannot detect any real novelty/contribution as such networks have been used before for medical image segmentation.
The evaluation is limited and does not compare to recent methods (also, there are no references in the main results table and some names there don't even appear in the reference list).